# A dietary sterol trade-off determines lifespan responses to dietary restriction in *Drosophila melanogaster* females

**Brooke Zanco, Christen K Mirth, Carla M Sgrò, Matthew DW Piper\***

Monash University, School of Biological Sciences, Clayton, Australia

**Abstract** Diet plays a significant role in maintaining lifelong health. In particular, lowering the dietary protein: carbohydrate ratio can improve lifespan. This has been interpreted as a direct effect of these macronutrients on physiology. Using *Drosophila melanogaster*, we show that the role of protein and carbohydrate on lifespan is indirect, acting by altering the partitioning of limiting amounts of dietary sterols between reproduction and lifespan. Shorter lifespans in flies fed on high protein: carbohydrate diets can be rescued by supplementing their food with cholesterol. Not only does this fundamentally alter the way we interpret the mechanisms of lifespan extension by dietary restriction, these data highlight the important principle that life histories can be affected by nutrient-dependent trade-offs that are indirect and independent of the nutrients (often macronutrients) that are the focus of study. This brings us closer to understanding the mechanistic basis of dietary restriction.

**\*For correspondence:**
matthew.piper@monash.edu

**Competing interests:** The authors declare that no competing interests exist.

## Introduction

Dietary restriction, also called calorie restriction, is a moderate reduction in food intake that extends healthy lifespan across a broad range of taxa, from yeast to primates (*Chapman and Partridge, 1996*; *Colman et al., 2009*; *Lin et al., 2002*; *McCay et al., 1935*). The generality of this observation has inspired confidence that the health benefits of dietary restriction might also be employed to improve human ageing (*Campisi et al., 2019*). In an attempt to harness its benefits, a great deal of current research is focused on discovering the nutritional components and the molecular mechanisms that underpin the lifespan benefits of dietary restriction (*López-Otín et al., 2013*; *Simpson et al., 2017*).

Our current understanding of how diet modifies lifespan has grown out of evolutionary theory and experiments using model organisms. The most prominent theoretical explanation has been the disposable soma theory, which employs resource-based trade-offs to explain how dietary restriction can benefit lifespan (*Kirkwood, 1977*; *Shanley and Kirkwood, 2000*). This theory postulates that organisms will maximise fitness by strategically partitioning limiting dietary energy either to reproduction or somatic maintenance, the latter determining lifespan. This means that longer lifespan is inevitably coupled with reduced reproduction because both traits compete for the same limiting resource.

Recent experimental work across a broad range of taxa has challenged the disposable soma theory by demonstrating that reproduction and lifespan respond predominantly to the balance of dietary macronutrients, not the overall energy content of the diet (*Mair et al., 2005*; *Lee et al., 2008*; *Skorupa et al., 2008*; *Grandison et al., 2009*; *Solon-Biet et al., 2014*; *Solon-Biet et al., 2015*; *Simpson et al., 2017*; *Regan et al., 2020*). Specifically, high protein, low carbohydrate diets are consistently associated with high reproduction and short lifespan, while low-protein, high-carbohydrate diets are associated with longer lifespan and lower levels of reproduction (*Piper et al., 2011*; *Simpson et al., 2017*). These data indicate that lifespan and reproduction are not in competition for

**eLife digest** For the past fifteen years, animal studies have consistently shown that a low-protein, high-carbohydrate ('carbs') diet can extend the lifespan of many organisms, but at the cost of the number of offspring an individual can produce. Yet, it is still unclear what the best dietary balance is, and how these effects arise. One potential explanation could be that reproduction damages the body: low levels of proteins would therefore prolong life by lowering the reproductive output.

Here, Zanco et al. examined the possibility that protein intake in fruit flies could instead be acting indirectly by changing the levels of a fat-like molecule called cholesterol, which is used to maintain the body and to support reproduction.

To test this idea, groups of fruit flies were fed high levels of proteins. This led to increased reproduction rates, in turn depleting the mothers' reserves of cholesterol. Without enough of the molecule in their diet, the insects were less able to maintain their bodies, which reduced their lifespan. When Zanco et al. added cholesterol to a high-protein diet, the flies lived for the normal length of time. Longer lifespan therefore did not require restriction of the diet or any of its components. In fact, the flies that lived the longest ate protein rich diets, and reproduced the most.

This study helps to better understand why changes in diet can influence how long an organism lives for, highlighting that the abundance of certain key molecules may be more important than restricting the levels of proteins, carbs or calories actually consumed.

limiting energy derived from the diet, but instead are optimised at different dietary protein: carbohydrate ratios. In response to these findings, an enormous effort is now focused on uncovering how macronutrient rebalancing, in particular protein dilution, acts to improve lifespan (*Blagosklonny, 2006*; *Blagosklonny, 2010*; *Moatt et al., 2020*; *Regan et al., 2020*; *Speakman, 2020*). Accumulating evidence indicates that the effect is mediated by reducing signalling through the amino acid sensitive Target Of Rapamycin (TOR) pathway to enhance cellular proteostasis (*Sanz et al., 2004*; *Ayala et al., 2007*; *Raubenheimer and Simpson, 2009*; *Simpson and Raubenheimer, 2009*; *Taylor and Dillin, 2011*; *Fanson et al., 2012*; *Sabatini, 2017*).

Although detrimental for lifespan, relatively high protein, low carbohydrate diets are beneficial for female reproduction (*Chong et al., 2004*; *Solon-Biet et al., 2015*). We have studied this closely in the fruit fly *Drosophila melanogaster*, where the principle driver of egg production is dietary protein (*Min and Tatar, 2006*; *Grandison et al., 2009*; *Piper et al., 2017*). Although protein is key, females must transfer dozens of nutrients into eggs for future embryo formation and not all of these components contribute to the flies' decision to produce eggs (*Piper et al., 2014*; *Mirth et al., 2019*; *Wu et al., 2020*). This means that high protein diets might drive mothers to produce eggs at a faster rate than they can support if the diet contains insufficient levels of the other components that are required to make eggs. In this scenario, the macronutrients would have an indirect effect on lifespan by changing the availability of another limiting nutrient that is required for somatic maintenance. If true, this would move the focus of mechanistic studies away from the direct effects of protein, TOR, and proteostasis, towards some other component of nutritional physiology. Distinguishing between these possible causes of death is important since it would fundamentally change our understanding of the way diet alters lifespan. It also has the important knock-on effect that we could change the way we design diets for longer life. For instance, supplementing high protein diets with key limiting nutrients would be as beneficial as restricting dietary protein or treating with pharmacological suppressors of TOR.

Of the many studies that have examined the effects of dietary protein and carbohydrate on lifespan and reproduction in Drosophila, most have done so by varying dietary yeast and sugar proportions, where yeast is the flies' natural source of protein (*Mair et al., 2005*; *Lee et al., 2008*; *Skorupa et al., 2008*). However, yeast also contains all of the flies' other essential macro and micronutrients whose relative proportions can change, and thus possibly interact with protein and carbohydrates to dictate life-history outcomes. We have previously found that depriving adult female flies of a source of sterols, an essential micronutrient for insects, imposes a minor cost on reproduction, but a substantial (>50%) cost to lifespan (*Piper et al., 2014*; *Wu et al., 2020*). These data indicate

that yeast sterol levels may contribute to the effects on lifespan of protein and carbohydrate. To investigate the interactions between dietary protein, carbohydrate, and sterols systematically, we have used the design principles of the geometric framework for nutrition (*Simpson and Raubenheimer, 2012*; *Simpson and Raubenheimer, 1993*) and a completely defined (holidic) diet that allows us to control the levels of each nutrient independently of all others (*Piper et al., 2014*; *Piper et al., 2017*). These data point to an important role for sterols in determining Drosophila lifespan, which we verified to be relevant in two yeast-based media that are often used in Drosophila lifespan studies. This work is critical to identifying how diet modifies lifespan at the molecular level, and highlights a new way to think about diet design to improve healthy ageing.

## Results

### Protein: carbohydrate ratio influences lifespan and reproduction

To examine the interactive effects of dietary protein, carbohydrate, and cholesterol on Drosophila lifespan and fecundity, we used our completely defined (holidic) diet (*Piper et al., 2014*) to manipulate each nutrient independently of all others. We selected dietary protein and carbohydrate concentrations that we know to elicit the full range of lifespan and fecundity responses to dietary restriction (*Lee et al., 2008*; *Piper et al., 2014*, *Piper et al., 2017*; *Ma et al., 2020*).

Similar to what we and others have found previously (*Mair et al., 2005*; *Lee et al., 2008*; *Grandison et al., 2009*; *Piper et al., 2014*, *Piper et al., 2017*; *Katewa et al., 2016*), lifespan and reproduction were modified by dietary protein manipulations (*Figure 1*). Specifically, egg production showed a linear, positive correlation with dietary protein content (*Figure 1*, *Supplementary file 1*), while lifespan showed a peak at intermediate protein (66 d median at 10.7 g/l), and fell away at both higher (49 d median at 33.1 g/l) and lower (43 d median at 5.2 g/l) concentrations (*Figure 1a–b*, *Supplementary file 2*). Thus, as is typical for dietary restriction experiments, restricting dietary protein from high to intermediate levels increased lifespan and decreased reproduction (*Lee et al., 2008*; *Skorupa et al., 2008*; *Grandison et al., 2009*; *Katewa et al., 2016*; *Le Couteur et al., 2016*).

When increasing dietary carbohydrate against an otherwise fixed nutritional background, egg laying was suppressed in a dose-dependent fashion, but lifespan remained at its maximum level and was unchanged across all carbohydrate doses (~66 d median, *Figure 1c–d*). The diet with the lowest concentration of carbohydrate (5.7 g/l), which also contained the intermediate protein level (10.7 g/l), supported both maximum lifespan (*Figure 1d*; 66 d median) and the highest level of egg laying (75 eggs/female) of any diet in our experiment. Thus, as we have previously shown (*Piper et al., 2017*), balancing the dietary protein and carbohydrate concentrations can reveal a single dietary optimum for both traits, showing that lifespan shortening is not necessarily caused by high egg laying alone.

### Cholesterol interacts with protein and carbohydrate to modify lifespan and reproduction

Most dietary restriction studies on Drosophila vary dietary protein by modifying the yeast levels in food (*Chapman and Partridge, 1996*; *Mair et al., 2005*; *Lee et al., 2008*; *Skorupa et al., 2008*). While yeast is the flies' major source of protein, it is also their only source of dozens of other nutrients, including sterols, which are essential micronutrients for insects (*Carvalho et al., 2010*). To quantify the effects of varying dietary sterol levels on fly lifespan and egg laying, we maintained flies on the same set of diets as above, varying in protein and carbohydrate concentrations, while also varying cholesterol across four different levels: 0 g/l, 0.15 g/l (low), 0.3 g/l (medium; also our standard level), and 0.6 g/l (high).

Initial analysis of the data showed that diet type, when defined by variation of macronutrient composition, as well as variation in cholesterol concentration both significantly modified egg laying and lifespan (*Figure 2*, *Supplementary files 3* and *4*). By contrast, we saw no main effect of dietary energy density (calories) on either trait, which is consistent with previous findings showing that these traits are driven by the relative proportion of protein and carbohydrate in the diet independently of caloric value (*Lee et al., 2008*; *Mair et al., 2005*; *Skorupa et al., 2008*). We therefore proceeded in our analysis to assess how cholesterol modified the effects of protein and carbohydrates on these traits.

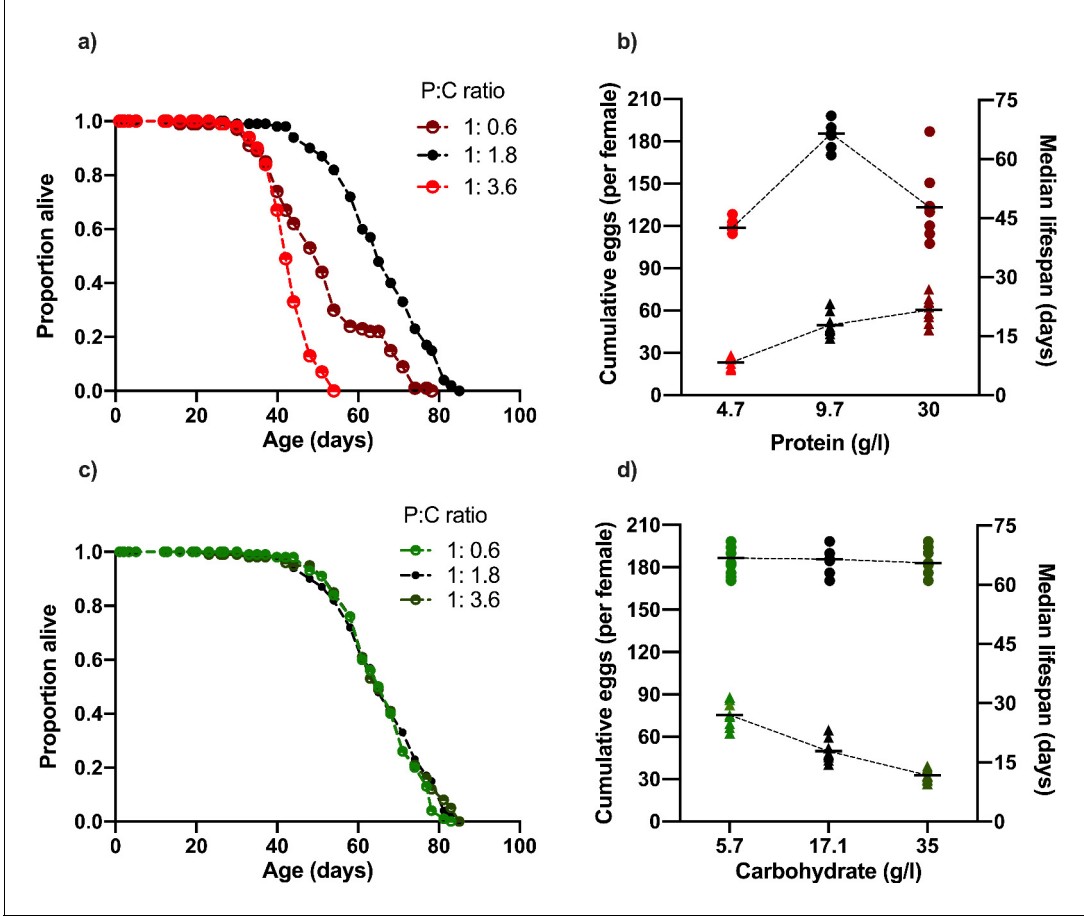

**Figure 1.** Changing dietary protein and carbohydrate concentrations modify Drosophila lifespan and fecundity. (**a, b**) Lifespan was maximised at our intermediate dose of dietary protein (carbohydrate fixed at 17.1 g/l) but was unaffected by our carbohydrate (**c, d**) concentration range (protein fixed at 9.7 g/l). (**b**) Cumulative egg production had a significant positive relationship with protein levels and (**d**) significant negative correlation with dietary carbohydrate content. Note that the intermediate protein and carbohydrate diet (9.7 g/l protein, 17.1 g/l carbohydrate) is common to both nutrient dilution series. The median survival data in panels (**b**) and (**d**) represent data from replicates that are combined in panels (**a**) and (**c**), respectively. (**b,d**) Cumulative eggs laid per female are represented as triangles while median survival (days) are shown as circles. Statistical analysis reported in *Supplementary files 1* and *2*.

We first compared the flies' responses to variation in both protein and cholesterol (*Figure 2—figure supplement 1a–b*). In general, lifespan was optimised at our intermediate dose of protein, while increasing cholesterol was beneficial, but with diminishing effect as its concentration was increased (*Figure 2*, *Supplementary file 5*). Interestingly, changing cholesterol modified the flies' lifespan response to protein, an effect that can be seen when the data are separated by level of cholesterol addition (*Figure 2—figure supplement 1a–b*). At 0 g/l cholesterol (*Figure 2a*) increasing protein concentration in the diet decreased lifespan. However, at 0.15 g/l cholesterol, the shape of the response changed such that only the highest protein concentration decreased lifespan (35 d median; *Figure 2c*) when compared with intermediate (9.7 g/l; 55 d median) and low-protein (4.7 g/l; 52 d median) diets. At 0.3 g/l of cholesterol, lifespan was highest on the diet with intermediate protein concentration (66d median) and flies on the high protein diet were longer lived (49 d median) than the flies on the lowest protein diet (43 d median). Finally, increasing cholesterol from 0.3 g/l to 0.6 g/l (*Figure 2g* – *Figure 2—figure supplement 1a–b*) did not change the way that lifespan responded to protein. Thus, lowering dietary cholesterol was detrimental for lifespan and it intensified the negative effects of increasing dietary protein concentrations.

Across the same set of diets, we observed a generally beneficial effect on egg laying of increasing dietary protein and cholesterol, and both had diminishing benefits as their concentrations increased (*Figure 2—figure supplement 1d–e*, *Supplementary file 6*). Cholesterol also modified the way egg

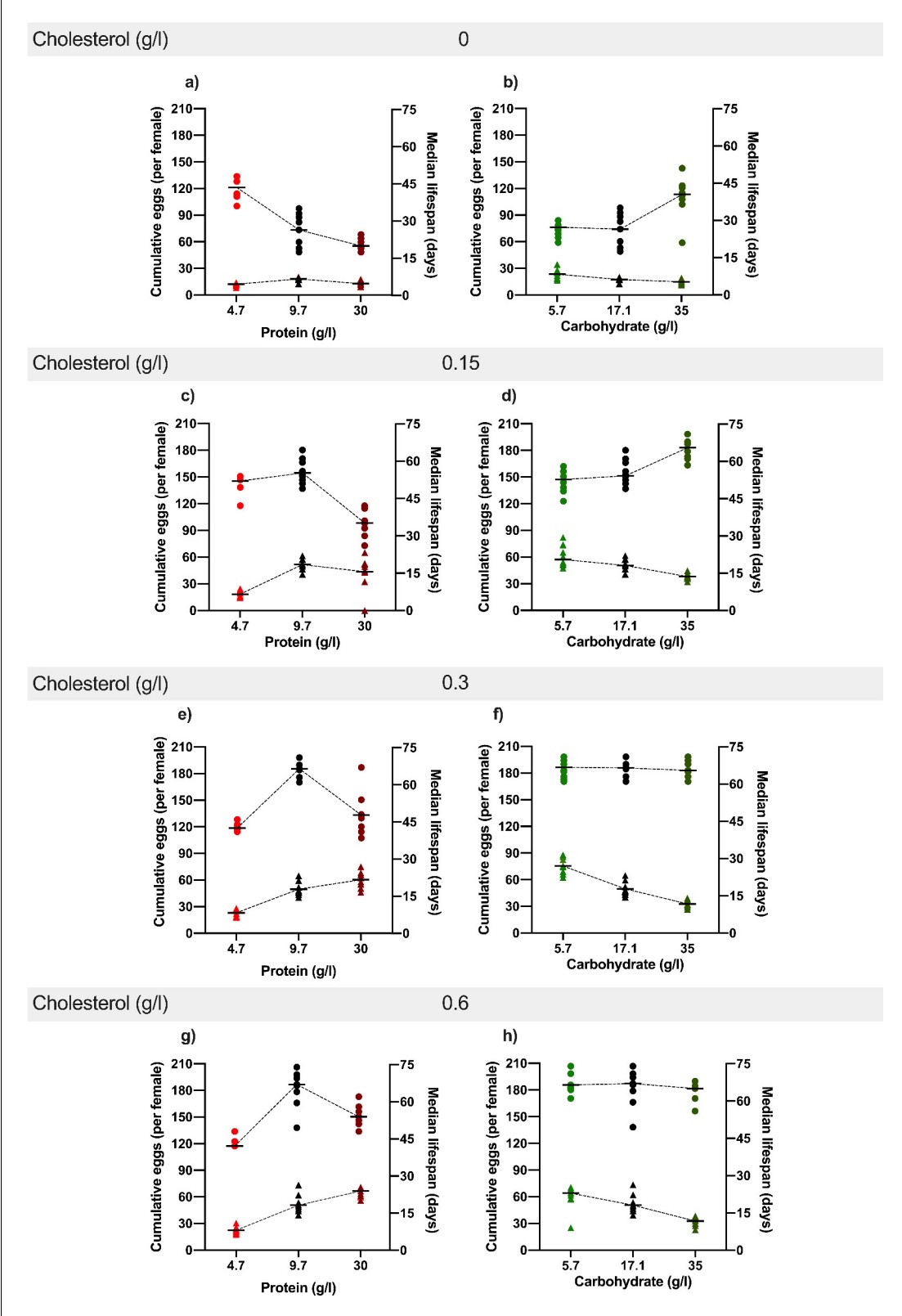

**Figure 2.** Dietary cholesterol content significantly modified the effect of protein and carbohydrate content on lifespan and reproduction. Lowering cholesterol most severely compromised lifespan as protein levels increased and as carbohydrate levels decreased. In general, increasing protein and decreasing carbohydrate drove increasing levels of egg production, and this was enhanced by increasing dietary cholesterol levels. Lines show the relationship between the cumulative eggs laid per female (left Y axis) or median survival (days) (right Y axis) and dietary protein or carbohydrate levels

*Figure 2 continued on next page*

*Figure 2 continued*

across each cholesterol level. (0 g/l (a, b), 0.15 g/l (c, d), 0.3 g/l (e, f), 0.6 g/l (g, h)). Cumulative eggs laid per female are represented as triangles while median survival (days) are shown as circles. Statistical analysis reported in *Supplementary files 5* and *6*.

The online version of this article includes the following figure supplement(s) for figure 2:

**Figure supplement 1.** Changing dietary protein and cholesterol concentrations modify Drosophila lifespan (**a–c**), while changing protein, cholesterol and carbohydrate concentrations modify egg production (**d–f**).

**Figure supplement 2.** Dietary energy intake (calories) does not mediate lifespan or egg production in Drosophila (**a, b**), while changing P:C ratio and cholesterol do (**a, b**).

laying was affected by dietary protein (*Figure 2*). Increasing cholesterol from 0 g/l (*Figure 2a*) to 0.15 g/l (*Figure 2c*) amplified the positive effect on egg laying of increasing dietary protein. Further increasing cholesterol to 0.3 g/l had an additional benefit for egg laying (*Figure 2e*), but only for flies on the highest protein diet (compare *Figure 2c* with *Figure 2e*), while increasing cholesterol even further, to 0.6 g/l (*Figure 2g*), did not change egg laying from that seen on 0.3 g/l. Thus, the response of egg laying to increasing protein was only compromised when cholesterol was completely removed from the diet, or when cholesterol was low (0.15 g/l) and protein was high (30 g/l) (*Figure 2c*).

Together, these data show that reducing cholesterol had negative effects on both lifespan and egg laying, and that these negative effects became more pronounced with increasing dietary protein. Furthermore, the negative interaction between lowering cholesterol and increasing protein was more severe and occurred at a lower protein concentration for lifespan than it did for egg laying.

Next, we looked to see if changing dietary cholesterol modified the responses of lifespan and egg laying to variation in carbohydrate concentration (*Figure 2b,d,f,h* - *Figure 2—figure supplement 1c and f Supplementary files 5* and *6*). At 0 g/l cholesterol, lifespan was generally short (31d median) but positively affected by increasing dietary carbohydrate (up to 40 d median) (*Figure 2b*). As dietary cholesterol was increased to 0.15 g/l, lifespan on all diets was higher and the positive effect of increasing carbohydrate was preserved (*Figure 2d*). However, when cholesterol reached 0.3 g/l, the flies were constantly long-lived, and lifespan was unaffected by dietary carbohydrate level (66 d median) (*Figure 2f*). This pattern was not changed by increasing cholesterol further to 0.6 g/l (*Figure 2h*). Thus, each of our dietary carbohydrate levels could support maximal fly lifespan, but the lower carbohydrate diets were more susceptible to the detrimental effects of dietary cholesterol dilution.

Increasing dietary carbohydrate had a generally negative impact on egg laying, and this effect was modified by the benefits of increasing dietary cholesterol (*Figure 2—figure supplement 1f*, *Supplementary file 6*). Without any cholesterol in the food, egg laying was consistently low and was negatively affected by increasing dietary carbohydrate (*Figure 2b*). This negative effect of carbohydrate on egg laying became stronger as cholesterol was increased to 0.15 g/l (*Figure 2d*) and 0.3 g/l (*Figure 2f*), with no further change as cholesterol increased from 0.3 g/l to 0.6 g/l (*Figure 2h*). This increasingly negative relationship between carbohydrate and egg laying was caused because increasing cholesterol benefited egg laying more at lower dietary carbohydrate levels – the opposite of what we observed for lifespan.

Thus, once again fly lifespan and egg laying worsened as cholesterol was diluted, but unlike its negative interaction with *increasing* dietary protein, the detrimental effects of lowering cholesterol became stronger as carbohydrate levels *decreased*. This indicates that the negative impact of lowering cholesterol is not a specific interaction with either high protein or low carbohydrate levels in the diet. We also found that the benefits of cholesterol addition were not related to the caloric content of the diet because cholesterol improved fecundity and extended lifespan of flies on almost all diets, including those with the lowest, intermediate, and highest caloric densities (*Figure 2—figure supplement 2*). Instead, lowering cholesterol produces worse outcomes as the dietary protein: carbohydrate ratio increases. This is the same change in macronutrient balance that promotes increasing egg laying.

## Increasing the dietary protein: carbohydrate ratio drives increasing reproduction and makes fly lifespan susceptible to dietary cholesterol dilution

We saw that flies produce more eggs in response to increasing dietary protein: carbohydrate ratio and that these positive effects persist even as dietary cholesterol falls to a level that cannot fully support lifespan (less than 0.3 g/l cholesterol). Thus, the protein: carbohydrate ratio appears to take precedence over dietary sterol levels in the decision to commit to reproduction. If this is the case, we expect to see a positive relationship between the dietary protein: carbohydrate ratio and egg laying across all levels of dietary cholesterol. This is indeed what we found, although the positive relationship was modified by cholesterol level (*Figure 3a,c,e,g*, *Supplementary file 7*), starting with a weak positive effect on 0 g/l cholesterol (*Figure 3a*) and becoming increasingly positive as cholesterol was increased to 0.15 g/l (*Figure 3c*) and 0.3 g/l (*Figure 3e*). Once again, increasing cholesterol from 0.3 g/l to 0.6 g/l promoted no further benefit (*Figure 3g*).

Reproduction can impose a cost on future survival if resources that are essential for somatic maintenance are preferentially committed to making eggs. Since increasing protein: carbohydrate levels drove increasing egg laying, even when the adults were completely deprived of sterols, it is possible that females are committing sterols to egg production at a rate faster than they can replenish it from the diet. If true, mothers on low cholesterol diets would become shorter lived as egg laying increases, but when cholesterol is sufficient, the relationship between egg production and lifespan should become less negative. To test this, we plotted egg laying against lifespan for all replicates across all diets. This showed that egg laying was a significant predictor of lifespan, and that this relationship was modified by dietary cholesterol (*Figure 3b,d,f,h*, *Supplementary file 8*). When the data are grouped by dietary cholesterol level (*Figure 3*), we see that when cholesterol was at 0 g/l (*Figure 3b*), there was a negative relationship between egg laying and lifespan, but as the cholesterol level increased, the correlation flattened, such that the slope was no longer negative for each level of cholesterol supplementation (*Figure 3d,f,h*, *Supplementary file 8*).

Thus, when dietary cholesterol was insufficient, increasing dietary protein: carbohydrate drove higher egg laying (*Figure 3a*) and this predicted lifespan shortening (*Figure 3b*) – a scenario that exemplifies the negative relationship between reproduction and lifespan in response to increasing protein: carbohydrate levels that is regularly reported in the dietary restriction literature (*Mair et al., 2005*; *Lee et al., 2008*; *Skorupa et al., 2008*; *Solon-Biet et al., 2014*; *Solon-Biet et al., 2015*; *Simpson et al., 2017*). However, when cholesterol was increased, the negative relationship was reduced such that egg laying was either completely independent of lifespan (*Figure 3d*) or became slightly positively correlated, indicating that the dietary conditions, which promote egg laying, are the same as those that promote longer lifespan (*Figure 3f,h*). Thus, higher egg laying in response to increasing protein: carbohydrate levels only shortens lifespan when cholesterol is insufficient to support egg production.

## Lifespan extension by rapamycin depends on dietary cholesterol level

TOR signalling is a key amino acid signalling pathway that is critical for growth, reproduction, and lifespan. Because TOR activity increases with dietary protein levels, it has been implicated as mediating the detrimental effects on lifespan of high protein diets (*Liu and Sabatini, 2020*). This is supported by the fact that rapamycin, a pharmacological suppressor of TOR, has been shown to extend lifespan across numerous species, including Drosophila where it also suppresses egg production across different caloric densities (*Bjedov et al., 2010*; *Schinaman et al., 2019*; *Scialò et al., 2015*). If sterol limitation is the reason why high egg production on high protein: carbohydrate diets causes reduced lifespan, rapamycin might extend lifespan because it reduces egg production and therefore rescues females from sterol depletion. If true, rapamycin should extend life only when the flies on high protein: carbohydrate diets are sterol limited.

As before, when we maintained flies on a high protein: carbohydrate diet, increasing dietary cholesterol from 0.1 to 0.3 g/l increased lifespan (62 d median v 69 d median)(*Figure 4a*). Egg laying was also slightly (34%), but significantly, elevated by cholesterol supplementation (*Figure 4b*) indicating that 0.1 g/l cholesterol was limiting for both lifespan and reproduction. When rapamycin was added to both foods, egg laying was almost completely suppressed (*Figure 4b*). Rapamycin also extended fly lifespan, but only for flies on low dietary cholesterol (0.1 g/l)(*Figure 4a*), bringing their

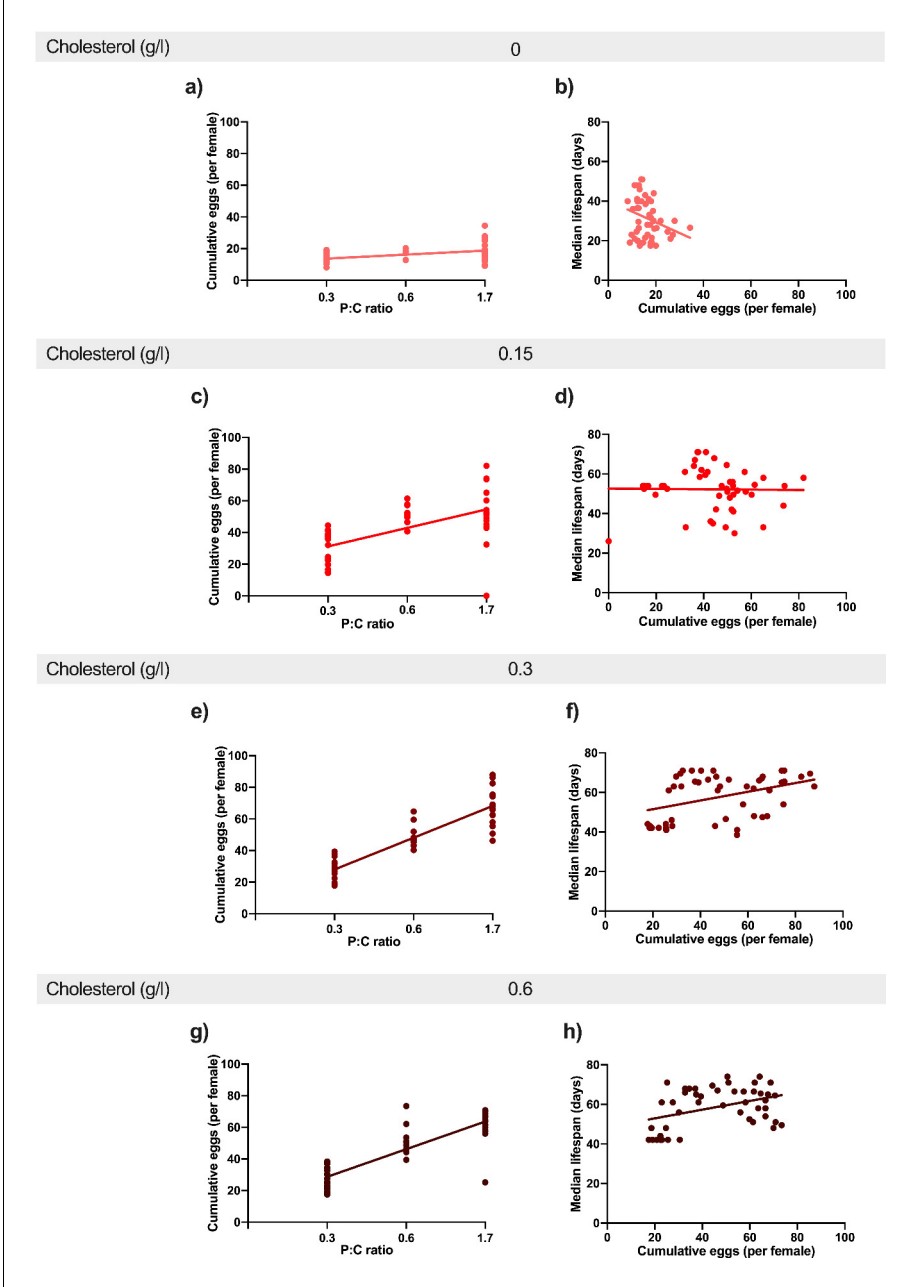

**Figure 3.** Increasing dietary protein: carbohydrate ratio resulted in increased egg production at every level of cholesterol. This positive effect was stronger from 0 g/l cholesterol (a) to 0.15 g/l (c) and 0.3 g/l (e). There was no additional benefit of further increasing cholesterol to 0.6 g/l (g). Regression lines show the relationship between cumulative eggs laid per female and protein: carbohydrate ratio. Providing adequate cholesterol transforms the relationship between egg production and lifespan from negative to positive. When cholesterol was not available (0 g/l) (b), there was a negative relationship between egg laying and lifespan as dietary protein and carbohydrate levels were varied. When cholesterol was provided at (0.15 g/l)(d) or above (f, h), this negative relationship was eliminated and egg production varied independently of lifespan. Regression lines show the relationship between cumulative eggs laid per female and median survival (days) in response to varying levels of cholesterol availability.

lifespan up to the same level as flies on higher cholesterol food (0.3 g/l; 69 d median). Adding rapamycin to the food with higher cholesterol did not result in any additional lifespan improvement over what was already achieved by increasing cholesterol alone (69 d median; *Figure 4a*).

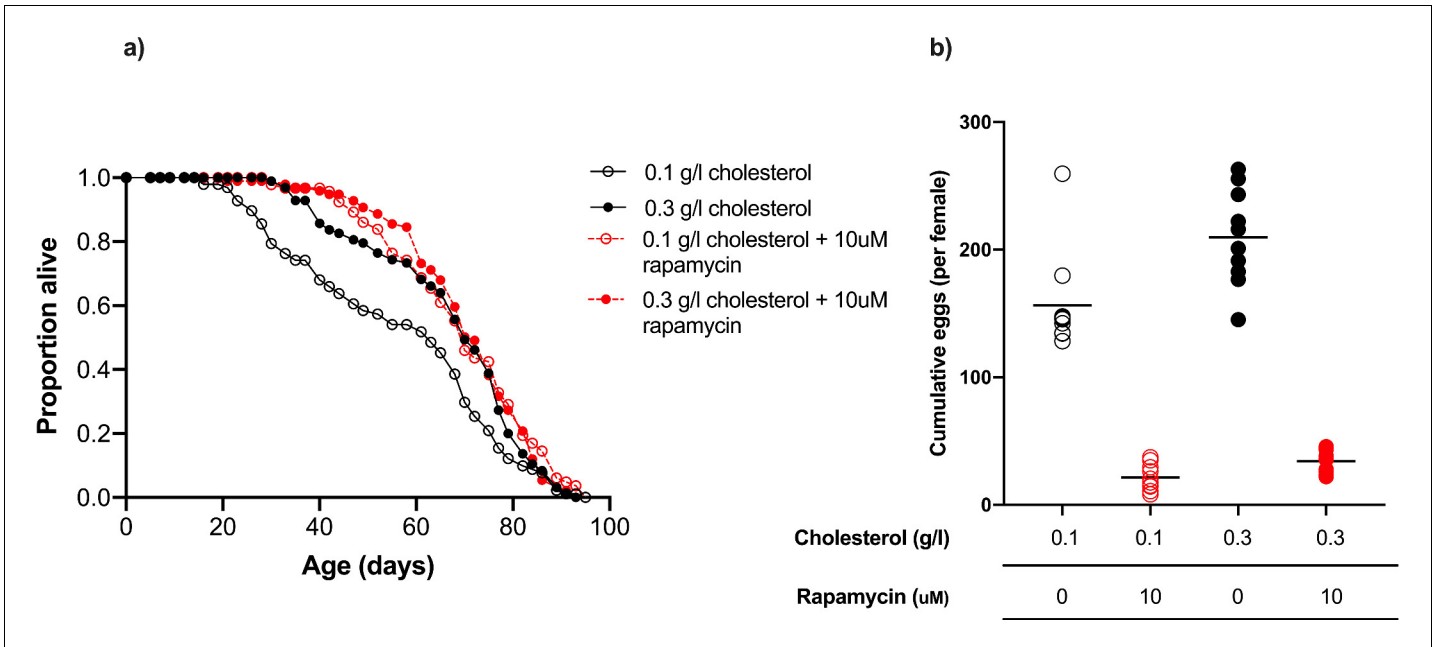

**Figure 4.** Rapamycin extends lifespan in flies consuming a low cholesterol diet (0.1 g/l) but had no effect when cholesterol level was increased to 0.3 g/l. (a) There was no significant difference in lifespan amongst flies fed 0.3 g/l cholesterol, 0.3 g/l cholesterol + rapamycin or 0.1 g/l cholesterol + rapamycin, all of which were significantly longer lived than flies fed 0.1 g/l cholesterol (0.1 g/l v 0.3 g/l, p=0.014; 0.1 g/l v 0.1 g/l + rapamycin, p<0.001; 0.1 g/l v 0.3 g/l + rapamycin, p=0.002, log rank test). (b) Cumulative eggs laid were significantly reduced in flies treated with rapamycin (p<0.001, two-way ANOVA), and also significantly reduced when cholesterol was limited (p<0.001, two-way ANOVA).

These data show that lifespan extension by rapamycin administration is conditional on the flies being on a low cholesterol diet. Together, our data are consistent with the flies' lifespan being determined by having access to sufficiently high levels of dietary sterols that they have enough left over after reproduction to meet their needs for somatic maintenance. This can be achieved either by enriching the amount of cholesterol in the diet, or by reducing the flies' expenditure on egg production, which can be achieved by reducing the dietary protein: carbohydrate ratio or by suppressing egg production pharmacologically.

## Standard yeast-based media used in the laboratory contains lifespan limiting levels of sterols

The experiments above were all performed using synthetic diets in which our ability to vary the absolute and relative concentrations of protein, carbohydrate, and sterol are limited only by solubility. However, most laboratories maintain fly populations on a diet that consists of yeast and sugar as well as variable numbers of other ingredients (*Piper, 2017*). Although the relative concentration of each nutrient in yeast is more constrained than on our synthetic diet, systematic studies have shown that the type and commercial source of yeast can have significant effects on overall dietary composition (*Lesperance and Broderick, 2020*) and the relationship between lifespan and egg laying (*Bass et al., 2007*). In *Bass et al., 2007*, the most dramatic lifespan reduction for increasing yeast was found when the fly food was made with a water-soluble extract of yeast that would contain very little, if any, sterols. Thus, similar to what we demonstrated on the synthetic diet, the shortening of fly lifespan when increasing the yeast content (protein: carbohydrate ratio) in lab foods may be caused by an insufficiency of dietary sterols.

We tested the effects of supplementing cholesterol into two sugar/yeast recipes that have been commonly used to study the effects of dietary restriction on lifespan (*Mair et al., 2005*; *Bass et al., 2007*; *Katewa et al., 2016*). These diets differ in both the number of ingredients used and the type of yeast; while both are *Saccharomyces cerevisiae*, one is a whole cell autolysate, while the other is a water-soluble extract. Adding 0.3 g/l cholesterol to both the low yeast (dietary restriction) and high yeast foods of both yeast types had a significant positive effect on lifespan (*Figure 5a,c*) and egg

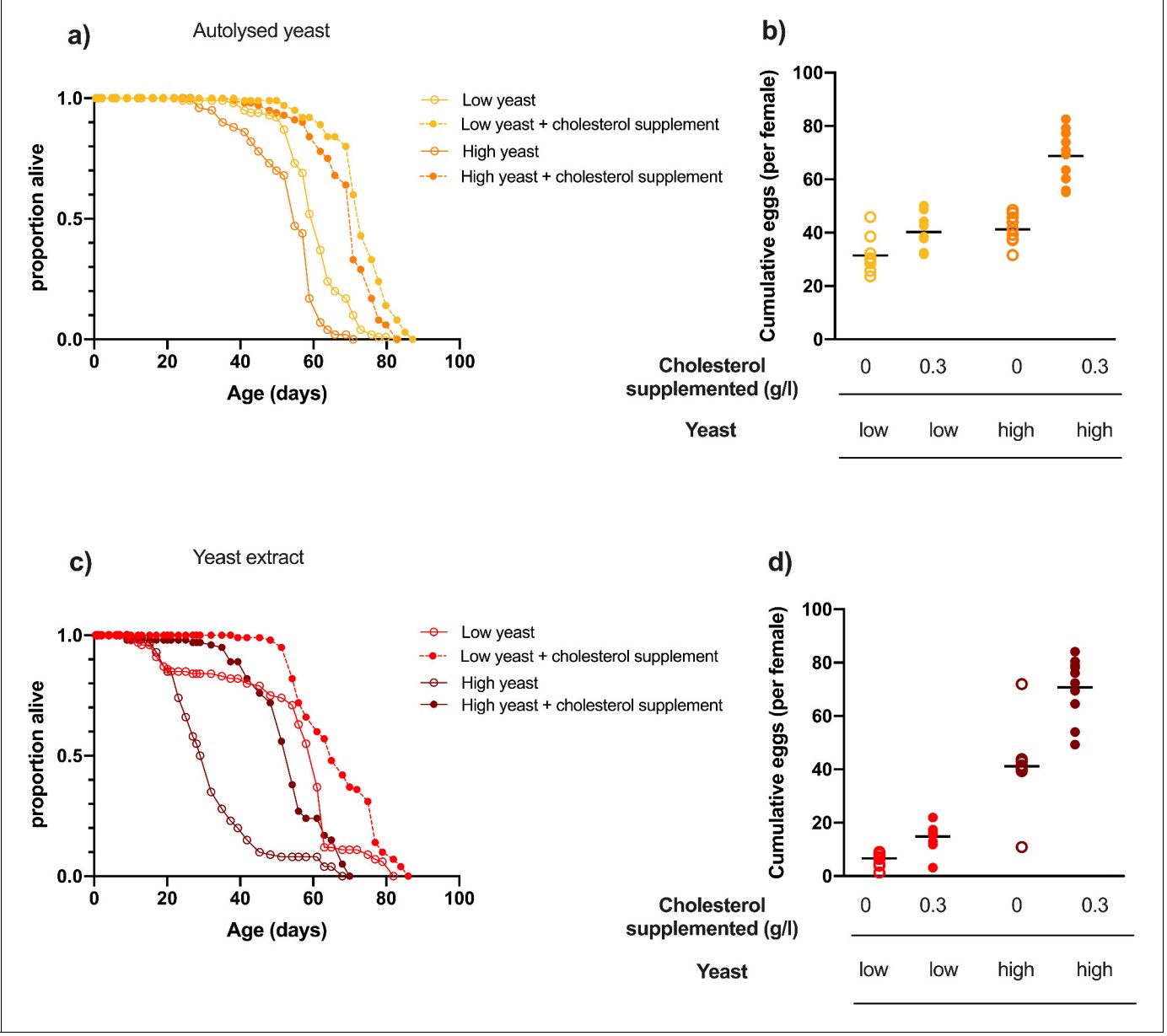

**Figure 5.** Cholesterol supplementation significantly extended lifespan and promoted egg laying of flies fed yeast-based diets. (**a**) Adding dietary cholesterol significantly increased the lifespan of flies on both high and low concentrations of diets made with autolysed yeast (low yeast v low yeast + cholesterol and high yeast v high yeast + cholesterol; p<0.001, log rank test). (**b**) Yeast and cholesterol addition to these two foods both positively affected egg production (p<0.001, two-way ANOVA). (**c**) Cholesterol addition significantly extended the lifespan of flies on diets made with yeast extract (low YE v low YE + cholesterol and high YE v high YE + cholesterol; p<0.001, log rank test). (**d**) Cumulative egg laying was also positively affected by yeast addition and cholesterol addition to each yeast level (p<0.001, two-way ANOVA).

laying (*Figure 5b,d*) when compared to diets without cholesterol supplementation. The magnitude of this benefit to lifespan was greater for flies on the high yeast foods than on the low yeast foods, meaning that cholesterol supplementation narrowed the difference between the dietary restriction vs high yeast diet from 9 to 4% for flies on the autolysed yeast diets (*Figure 5a*) and from 81 to 25% lifespan extension for flies on the yeast extract diets (*Figure 5c*). We note that even with cholesterol supplementation, the flies on the high yeast diet were still significantly shorter lived than those on the cholesterol supplemented low yeast food. This small additional cost of the high yeast food could reflect a detrimental (toxic) effect on lifespan of very high dietary protein, similar to what we

observed in our highest protein diets on the synthetic foods (*Figure 2e,g*). This is not rescuable by cholesterol supplementation and is not related to the number of eggs that females produce.

## Discussion

The reason that higher protein: carbohydrate diets shorten lifespan in dietary restriction studies is routinely attributed to their direct effects on nutrient signalling pathways and physiology. However, our data implicate a fundamentally different mechanism in which the macronutrients act indirectly, by manipulating sterol availability, which then modifies lifespan. Specifically, diets with high protein: carbohydrate ratios decrease lifespan by causing mothers to overinvest limiting sterols into egg production. Thus, although the macronutrients set egg laying rates, it is actually the sterols that determine lifespan due to a trade-off with reproduction. The corollary of this finding is that the lifespan of flies on high protein: carbohydrate diets can be extended by increasing the supply of cholesterol. This approach is the opposite of, but complementary to, the already recognised strategies to extend lifespan by dietary restriction, which reduce maternal investment into reproduction by decreasing the dietary protein: carbohydrate ratio (*Mair et al., 2005*; *Lee et al., 2008*; *Skorupa et al., 2008*) or by treating the animals with rapamycin that suppresses TOR and reduces reproduction (*Bjedov et al., 2010*; *Harrison et al., 2009*; *Liu and Sabatini, 2020*). It is also consistent with our recent work that showed non-reproducing adult males and genetically sterile females suffer little to no lifespan cost when sterol deprived, which is presumably because they conserve sterols which would otherwise be depleted by reproduction (*Wu et al., 2020*).

### High protein diets promote egg production, driving a lethal micronutrient deficiency

In the lab, flies can be successfully reared and maintained on a mixture of just sugar and yeast (*Pearl and Parker, 1921*). This diet is thought to reflect their natural diet of rotting fruit and the microbial community – principally the yeasts – that cause the fruit to decay (*Markow et al., 2015*; *Piper, 2017*). Yeast contains all of the nutrients that flies require, including protein (~45%), carbohydrate (~40%), a small amount of fat (~7%), nucleic acids (~7%), and micronutrients, such as sterols, metal ions and vitamins, which are essential for flies. Drosophila rely heavily on protein from yeast, as well as carbohydrate from both yeast and plant sources, to guide their feeding behaviour. They select amongst foods containing the appropriate protein and carbohydrate concentrations to enhance their fitness (*Ribeiro and Dickson, 2010*; *Vargas et al., 2010*; *Walker et al., 2017*). Many of the other nutrients from their diet, including sterols, do not affect feeding behaviour, presumably because they are normally acquired in adequate quantities as part of a diet that is sufficient in macronutrients (*Walker et al., 2015*; *Leitão-Gonçalves et al., 2017*; *Münch et al., 2020*).

While the proportion of protein and carbohydrate in yeast remains relatively constant across growth conditions, the abundance of sterols can vary over a 10-fold range in response to changes in oxygen availability, which is essential for sterol biosynthesis (*Starr and Parks, 1962*; *Wilson and McLeod, 1976*). Thus, because fly feeding behaviour and egg production are almost entirely shaped by the macronutrients, fly lifespan is susceptible to reductions in the sterol: protein content of dietary yeast. Our data indicate that this is because protein drives sterols to be preferentially partitioned towards reproduction at the expense of maintaining the adult soma. While we have found this to be the case for flies feeding on lab based foods, it is also reasonable to expect it for flies feeding on rotting fruit, where microbial growth is largely fermentative (driven by high sugar levels and limiting oxygen), producing ethanol and short chain acids to which Drosophila has evolved a healthy tolerance (*Geer et al., 1993*).

### Extending fly lifespan by dietary restriction involves an indirect trade-off

There have been several theoretical attempts to describe the mechanistic basis for the lifespan benefits of dietary restriction (*Blagosklonny, 2006*; *Blagosklonny, 2010*; *Kirkwood and Rose, 1991*; *Moatt et al., 2020*; *Regan et al., 2020*; *Speakman, 2020*). In particular, the disposable soma theory proposes that organisms will strategically reallocate nutrients towards somatic maintenance at the cost of reproduction when nutrients are scarce and that this enhances lifespan (*Kirkwood and Rose, 1991*). Our data indicate that this trade-off can exist for flies feeding on yeast, but only when dietary

sterols are limiting. However, when dietary sterols are not limiting, this trade-off does not need to exist and a single nutritional optimum for both lifespan and reproduction can be found. Thus, the macronutrient balance that drives higher egg laying does not necessarily inflict a direct cost on lifespan.

In mechanistic work, the increased lifespan under dietary restriction has been attributed to the benefits of reduced dietary protein, which enhances proteome maintenance via reduced TOR signalling (*Harrison et al., 2009*; *Partridge et al., 2011*; *Kapahi et al., 2017*; *Piper et al., 2017*; *Sabatini, 2017*; *Dobson et al., 2018*; *Liu and Sabatini, 2020*). Interestingly, lysosomal cholesterol levels have recently been found to be a potent modifier of mTORC1 activity (*Castellano et al., 2017*; *Zhang et al., 2020*), which raises the possibility that both protein depletion and cholesterol addition modify ageing by reducing TOR signalling. However, the published data shows that cholesterol is an activator of TOR and cholesterol depletion inhibits its activity. Thus, we expect adding cholesterol to the diet would not reduce TOR signalling, but instead optimise conditions for maximal TOR signalling – especially on the high protein: carbohydrate diets in which we find cholesterol addition to be most effective for prolonging lifespan. These data indicate, therefore, that long life is possible when TOR signalling is high as long as the flies have sufficient sterols in their diets. Alternatively, longevity can still be assured under sterol-limiting conditions by reducing the cost of reproduction, either by reducing dietary protein, by adding rapamycin which suppresses reproduction or by making flies infertile (*Wu et al., 2020*). These data indicate that longevity assurance in *D. melanogaster* is not the result of enhanced proteostasis triggered by lowered TOR, but is instead, a side effect of avoiding sterol depletion caused by an over-investment in egg production.

Rapamycin is known to extend the lifespan of various organisms including *C. elegans*, yeast and mammals (*Harrison et al., 2009*; *Kapahi et al., 2010*; *Powers et al., 2006*; *Robida-Stubbs et al., 2012*). Because *C. elegans* cannot synthesise its own sterols, rapamycin might increase lifespan by preventing sterol depletion in a manner similar to what we have observed in *Drosophila*. However, sterols may not be lifespan limiting in other organisms such as yeast and mammals that have the ability to synthesise their own sterols. One explanation is that the administration of rapamycin prevents other micronutrient deficiencies caused by over-investment in growth and/or reproduction in response to high levels of dietary protein. For instance, rodents will export calcium from their own bones and teeth to meet the demands of pregnancy and lactation (*Miller and Bowman, 2004*; *Ozbek et al., 2004*; *Speakman, 2008*). For this reason, it would be interesting to see if providing additional micronutrients to the diets of ad libitum-fed mice can mimic the benefits of dietary restriction, similar to what we see for sterol supplementation in flies.

Another possibility is that rapamycin rescues animals from the effects of protein toxicity, which can occur at concentrations of protein higher than what we used in this study. In our experiments, we limit the upper range of dietary protein concentrations so as not to exceed those that are beneficial to reproduction. This practice is informed by the desire to study how dietary restriction enhances somatic maintenance to extend life in already healthy individuals, rather than studying the reduced risk of dying that occurs when flies are prevented from over-consuming. To test this, it would be interesting to study the effects of rapamycin addition over a broader range of protein concentrations than what we have used. If true, this has the important implication that rapamycin, and indeed different diet compositions, may prolong animal lifespan by more than one molecular mechanism. These are important implications to explore since the majority of work studying ageing in lab organisms is based on the assumption that the mechanisms are evolutionarily conserved, in the hope that they will benefit humans.

## Conclusion

Our data show that the detrimental effects of a high protein: carbohydrate diet on lifespan in female *Drosophila melanogaster* are, to a significant extent, driven by an indirect nutrient trade-off, in which the macronutrients drive maternal sterol depletion by enhancing egg laying. This is a fundamentally different mechanism from the predominant view that reducing protein: carbohydrate balance in diets improves lifespan by a direct action to reduce TOR signalling and enhance proteostasis. Because of our discovery, we show that the shortened lifespan of flies on a high protein: carbohydrate diet can be improved by supplementing their diet with cholesterol, as well as by reducing egg production by lowering the dietary protein: carbohydrate ratio or by administering rapamycin. Further work is now needed to discover the mechanisms through which cholesterol works to modify

lifespan in *Drosophila melanogaster*, and the role of other important micronutrients in healthy ageing across taxa.

## Materials and methods

### Key resources table

| Reagent type (species) or resource | Designation | Source or reference | Identifiers | Additional information |
|---|---|---|---|---|
| Chemical compound, drug | Rapamycin (Sirolimus) | Jomar Life Research | S1039 | |

### Fly husbandry

All experiments were conducted using a wild type *Drosophila melanogaster* strain called Dahomey (*Mair et al., 2005*).These flies have been maintained in large numbers with overlapping generations to maintain genetic diversity. Upon removal from their population cages, flies were reared for two generations at a controlled density before use in experiments, to control for possible parental effects. Eggs for age-synchronised flies were collected over 18 hr, and the resulting adult flies emerged during a 12 hr window. They were then allowed to mate for 48 hr before being anaesthetised with $CO_2$, at which point females were separated and allocated into experimental vials. Stocks were maintained and experiments were conducted at 25 °C on a 12 hr: 12 hr light:dark cycle at 65% humidity (*Bass et al., 2007*).

### Lifespan assays

For all lifespan assays, flies were placed into vials ( FS32, Pathtech) containing 3 ml of treatment food at a density of ten flies per vial, with ten replicate vials per treatment. Flies were transferred to

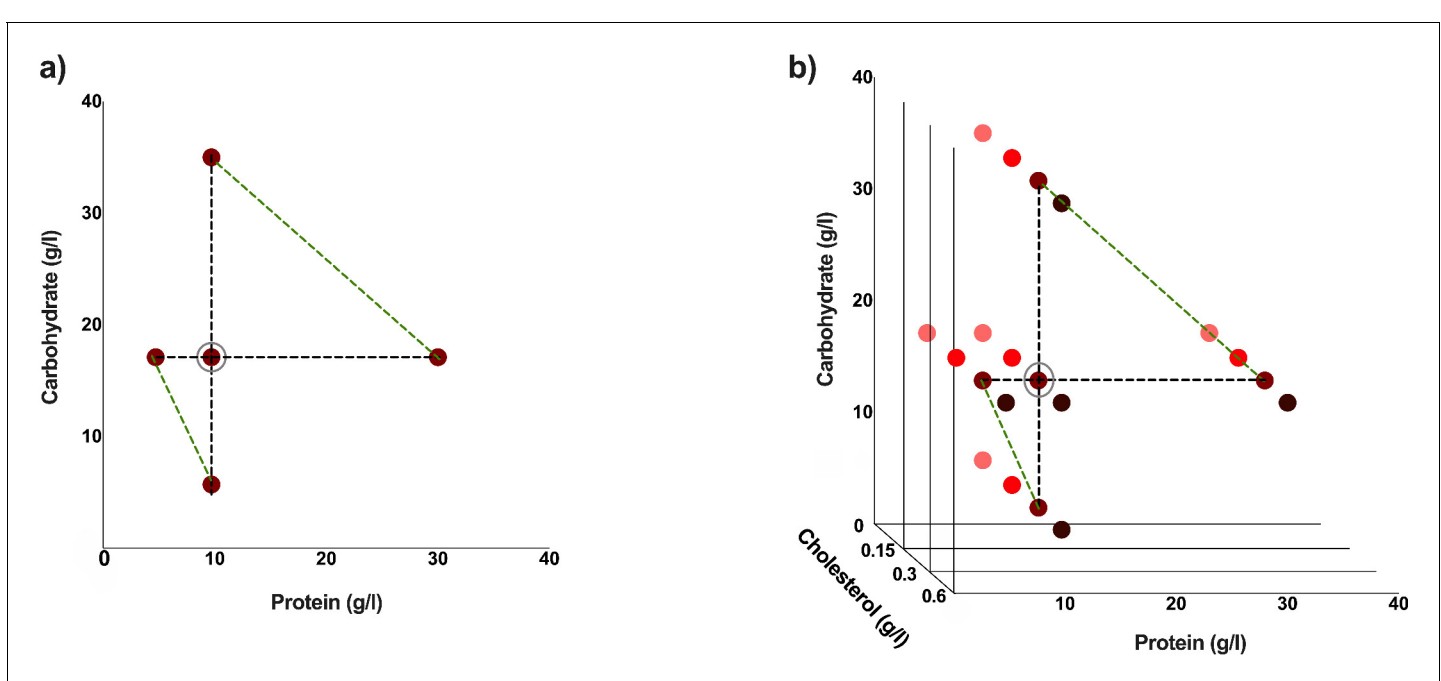

**Figure 6.** Experimental diets used are indicated by coloured dots. These diets have varying protein: carbohydrate ratios. This makes a total of five different experimental diets (**a**). The standard cholesterol concentration is 0.3 g/l. Three additional cholesterol concentrations were used for each of the five protein: carbohydrate ratios to make a total of 20 different experimental diets (**b**). Diets which are either protein constant or carbohydrate constant are connected by black dotted lines, and diets with comparable caloric concentrations are connected by green-dotted lines (**a, b**). The standard diet used in our laboratory is circled in grey (a, b).

**Table 1.** Protein: carbohydrate ratio, along with the nutrient densities, cholesterol concentration and caloric content, for all synthetic experimental diets used.

In the holidic media, amino acids are used to make up protein equivalents. To convert amino acids to protein equivalents, we used the molar quantities of nitrogen and the assumption that N makes up 16% of whole proteins (*Imafidon and Sosulski, 1990*). Calories were calculated using the method described in *Southgate and Durnin, 1970*.

| Diet | Protein: carbohydrate equivalent | Sum mass of amino acids (g/l) | Equivalent protein (g/l) | Carbohydrate (g/l)[†] | Cholesterol (g/l) | Estimated caloric content (kcal/l) |
|---|---|---|---|---|---|---|
| 1 | 1:3.6 | 5.25 | 4.7 | 17.1 | 0 | 87.2 |
| 2 | 1:3.6 | 5.25 | 4.7 | 17.1 | 0.15 | 87.2 |
| 3 | 1:3.6 | 5.25 | 4.7 | 17.1 | 0.3 | 87.2 |
| 4 | 1:3.6 | 5.25 | 4.7 | 17.1 | 0.6 | 87.2 |
| 5 | 1:3.6 | 10.74 | 9.7 | 35 | 0 | 178.8 |
| 6 | 1:3.6 | 10.74 | 9.7 | 35 | 0.15 | 178.8 |
| 7 | 1:3.6 | 10.74 | 9.7 | 35 | 0.3 | 178.8 |
| 8 | 1:3.6 | 10.74 | 9.7 | 35 | 0.6 | 178.8 |
| 9 | 1:1.8 | 10.74 | 9.7 | 17.1 | 0 | 107.2 |
| 10 | 1:1.8 | 10.74 | 9.7 | 17.1 | 0.15 | 107.2 |
| 11[*] | 1:1.8 | 10.74 | 9.7 | 17.1 | 0.3 | 107.2 |
| 12 | 1:1.8 | 10.74 | 9.7 | 17.1 | 0.6 | 107.2 |
| 13 | 1:0.6 | 33.1 | 30 | 17.1 | 0 | 188.4 |
| 14 | 1:0.6 | 33.1 | 30 | 17.1 | 0.15 | 188.4 |
| 15 | 1:0.6 | 33.1 | 30 | 17.1 | 0.3 | 188.4 |
| 16 | 1:0.6 | 33.1 | 30 | 17.1 | 0.6 | 188.4 |
| 17 | 1:0.6 | 10.74 | 9.7 | 5.7 | 0 | 61.6 |
| 18 | 1:0.6 | 10.74 | 9.7 | 5.7 | 0.15 | 61.6 |
| 19 | 1:0.6 | 10.74 | 9.7 | 5.7 | 0.3 | 61.6 |
| 20 | 1:0.6 | 10.74 | 9.7 | 5.7 | 0.6 | 61.6 |

* Standard diet.

† Carbohydrate is added to the diet as sucrose.

fresh vials every two to three days at which point deaths and censors were recorded and saved using the software package Dlife (*Linford et al., 2013*; *Piper and Partridge, 2016*).

## Fecundity assays

Fecundity was measured as the sum of the mean number of eggs laid per female once per week over four weeks (commencing on approximately day 8 of the experiment), except for the sugar yeast (SY) medium experiment, for which egg counts were recorded in weeks one, two and three. These timepoints were selected because measuring reproductive output during the first weeks of egg laying has shown to be representative of life-long fecundity in flies (*Chapman and Partridge, 1996*). The eggs laid on the food surfaces of all vials were imaged using a web camera mounted on a Zeiss dissecting microscope and eggs were counted both manually and using Quantifly (*Waithe et al., 2015*). Quantifly was trained using five images for each cholesterol concentration due to variance in food opacity.

## Experimental Diets

### Holidic medium experiments

To examine the effects of protein: carbohydrate ratio on lifespan and fecundity we chose five experimental diets that consisted of three different protein (amino acid): carbohydrate (sucrose) ratios at three levels of similar caloric densities (*Figure 6a*, *Table 1*). These diets also made up a three-diet series of protein only dilution, and a three-diet series of carbohydrate only dilution (*Figure 6a*). The

two diet series had one diet in common, which was our most commonly used, 'standard' lab diet (*Piper et al., 2014*). These diets incorporate those known to maximise either lifespan, reproduction or both (*Ma et al., 2020*; *Piper et al., 2017*). To examine the effects of cholesterol on these traits, we selected four cholesterol concentrations for each of these five diets, making a total of 20 diets (*Figure 6b*, *Table 1*). All diets were made using the holidic medium described in *Piper et al., 2014*, in which free amino acids are used to make up protein equivalents. To convert amino acids to protein equivalents, we used the molar quantities of nitrogen and the assumption that N makes up 16% of whole proteins (*Sosulski and Imafidon, 1990*). In this case, an amino acid ratio matched to the exome of adult flies (Flyaa) was utilised (*Supplementary file 9*; *Ma et al., 2020*; *Piper et al., 2017*). Finally, for practical reasons we used cholesterol in the diet as opposed to ergosterol, because it is easily accessible, and where studied, has been shown to be adequate to support Drosophila adult nutrition to the same extend as a yeast-based diet (*Piper et al., 2014*).

## Rapamycin experiment

The same methods for making the holidic medium described above were used to make all diets used in the rapamycin experiment. In this case however 18.9 g/l protein: 17.1 g/l carbohydrate were used. Cholesterol was added to the diet at a concentration of either 0.1 g/l or 0.3 g/l (cholesterol supplemented) and rapamycin was added to a final concentration in the diet of 10 μM. Diets were either un-supplemented, supplemented with cholesterol, rapamycin, or both.

## Yeast based experiments

Four sugar/yeast (SY) diets were created using sucrose (Bundaberg Sugar, Melbourne Distributors) and either whole yeast autolysate (MP Biomedicals, LLC, #903312) or yeast extract (Bacto Yeast Extract, #212750). These diets correspond to previously published conditions for high protein (fully fed) and low protein (dietary restriction) conditions (*Bass et al., 2007*; *Katewa et al., 2016*; *Mair et al., 2005*). The high protein diets contained, per litre 50 g sucrose and 200 g autolysed yeast, or 50 g sucrose, 50 g yeast extract plus 86 g of cornmeal (The Full Pantry, Victoria, Australia). The low- protein diets contained, per litre 50 g sucrose and 100 g autolysed yeast or 50 g sucrose, 5 g yeast extract plus 86 g cornmeal. To each of these diets, we added cholesterol (Glentham Life Sciences, GEO100, #100IEZ) at a concentration of either 0 or 0.3 g/l. Cholesterol was added to all diets as a powder which was mixed in with all other dry ingredients prior to cooking. This gave us a total of four experimental diets per yeast.

## Statistical analyses

All statistical analyses were performed using R (version 3.3.0, available from http://www.R-project.org/). One outlier was removed from the data set as the total number of eggs laid for that particular vial was more than two standard deviations from the mean. Omitting this point did not modify the significance of any of the statistical analyses or change any conclusions. For each experimental vial the median lifespan and mean number of eggs laid were obtained prior to analysis. Linear mixed effect models were used to analyse all data obtained using the holidic media. For the analysis of data obtained using the holidic media, a model reduction was performed by stepwise removal of the most complex non-significant term until any further removal significantly reduced the model fit. Log rank tests were used to compare the survival curves in the rapamycin experiment and yeast based dietary experiments. Finally, two-way ANOVAs were used to analyse egg laying results for the yeast based experiments and rapamycin experiment. Plots were made in Graphpad Prism (version 8.4.2).

## Acknowledgements

We would like to thank Amy Dedman from Monash University for technical assistance, Xiaoli He and Dr. Mingyao Yang (University College London at the time) for contributions to the early phase of these experiments, as well as Lisa Rapley (Monash University) for help with egg counts.

## Additional information

### Funding

| Funder | Grant reference number | Author |
|---|---|---|
| Australian Research Council | FT150100237 | Matthew DW Piper |
| National Health and Medical Research Council | APP1182330 | Matthew DW Piper |
| Australian Research Council | | Carla M Sgrò |
| Monash University | | Carla M Sgrò |
| Australian Research Council | FT170100259 | Christen K Mirth |

The funders had no role in study design, data collection and interpretation, or the decision to submit the work for publication.

### Author contributions

Brooke Zanco, Conceptualization, Data curation, Formal analysis, Investigation, Visualization, Methodology, Writing - original draft; Christen K Mirth, Conceptualization, Formal analysis, Supervision, Writing - review and editing; Carla M Sgrò, Conceptualization, Supervision, Writing - review and editing; Matthew DW Piper, Conceptualization, Formal analysis, Supervision, Funding acquisition, Methodology, Project administration, Writing - review and editing

### Author ORCIDs

Brooke Zanco (iD) https://orcid.org/0000-0001-7112-8640
Christen K Mirth (iD) http://orcid.org/0000-0002-9765-4021
Matthew DW Piper (iD) https://orcid.org/0000-0003-3245-7219

### Decision letter and Author response

Decision letter https://doi.org/10.7554/eLife.62335.sa1
Author response https://doi.org/10.7554/eLife.62335.sa2

## Additional files

### Supplementary files

• Supplementary file 1. Estimates from a linear mixed effects model to explain the effects of protein and carbohydrate on cumulative eggs laid per female, with replicate as a random effect. Decreasing doses of carbohydrate and increasing doses of protein resulted in significantly increased egg production.

• Supplementary file 2. Estimates from a linear mixed effects model to explain the effects of protein and carbohydrate on lifespan (median lifespan (days)), with vial as a random effect. While variations in carbohydrate had no effect on lifespan, increasing doses of protein resulted in a significant change in lifespan. Visual inspection of the data agreed with our past experience with these diets (*Piper et al., 2014*; *Piper, 2017*) that the lifespan response was best modelled by the quadratic term for protein (Protein$^2$) since lifespan peaked at our intermediate protein dose and fell away at both higher and lower doses. The quadratic term for Carbohydrate was thus also added to maintain balance in the model (Carbohydrate$^2$). The interaction between protein and carbohydrate is not included in any of our analyses because these terms were not co-varied in a balanced way in our experimental design.

• Supplementary file 3. Effects on median lifespan (days) of calories, cholesterol and diet type (a categorical variable indicating if either protein or carbohydrate was varied). Cholesterol had a significant positive effect on median lifespan, while diet type had a significant effect on median lifespan. Calories had no significant effect on lifespan. Data were analysed using a linear model with mixed effects, with vial as a random effect.

• Supplementary file 4. Effects on cumulative eggs per female of calories, cholesterol and diet type (a categorical variable indicating if either protein or carbohydrate was varied). Cholesterol had a significant positive effect on cumulative eggs per female, while diet type had a significant effect on cumulative eggs per female. Calories had no significant effect on cumulative eggs per female. Data were analysed using a linear model with mixed effects, with vial as a random effect.

• Supplementary file 5. Minimum adequate model describing the effects of protein, protein$^2$ carbohydrate, carbohydrate$^2$, cholesterol, cholesterol$^2$ and, where appropriate, their interactive effects on median lifespan (days). Data were analysed using a linear model with mixed effects, with vial as a random effect.

• Supplementary file 6. Minimum adequate model describing the effects of protein, protein$^2$ carbohydrate, carbohydrate$^2$, cholesterol, cholesterol$^2$ and, where appropriate, their interactive effects on cumulative eggs per female. Data were analysed using a linear model with mixed effects, with vial as a random effect.

• Supplementary file 7. Effects on cumulative eggs per female of P:C ratio, cholesterol, cholesterol$^2$ and the interaction between P:C ratio and cholesterol. Each of the main effects had a significant positive effect on egg production, and the amount of cholesterol significantly modified how P:C affected egg laying. Data were analysed using a linear model with mixed effects, with vial as a random effect.

• Supplementary file 8. Effects on median lifespan (days) of cumulative eggs per female, cholesterol, cholesterol$^2$ and the interaction between cumulative egg production and cholesterol. Cumulative egg production and cholesterol had a significant positive effect on median lifespan, while cholesterol$^2$ had a significant negative effect on median lifespan. Data were analysed using a linear model with mixed effects, with vial as a random effect.

• Supplementary file 9. The relative proportions of each amino acid in the FLYaa amino acid mixture used in this study.

• Transparent reporting form

## Data availability

All data gathered during this study is included in the manuscript and all datasets have been published on Figshare at the following URL: https://doi.org/10.26180/5f4d0cf8bb103.

The following dataset was generated:

| Author(s) | Year | Dataset title | Dataset URL | Database and Identifier |
|---|---|---|---|---|
| Zanco B, Mirth CK, Sgrò CM, Piper MD | 2020 | Survival & egg laying data for Zanco et al, 2020 | https://doi.org/10.26180/5f4d0cf8bb103 | figshare, 10.26180/5f4d0cf8bb103 |

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
