## [Decision Letter]

**Acceptance summary:**

This is a nicely conceived and written work, which convincingly shows the key role of dietary sterols as limiting tradeoff/coupling units between reproduction and somatic maintenance when protein : carbohydrate ratios increase. This work also sheds light on how rapamycin impacts female fly lifespan on a low sterol diet and high protein:carbohydrate ratios by indirectly preventing somatic sterol depletion via suppression of reproduction.

**Decision letter after peer review:**

Thank you for submitting your article "A dietary sterol trade off determines lifespan responses to dietary restriction in *Drosophila melanogaster*" for consideration by *eLife*. Your article has been reviewed by three peer reviewers, including Dario Riccardo Valenzano as the Reviewing Editor and Reviewer #1, and the evaluation has been overseen by Jessica Tyler as the Senior Editor.

The reviewers have discussed the reviews with one another and the Reviewing Editor has drafted this decision to help you prepare a revised submission.

Zanco et al. study how dietary sterols affect the fly responses to varying carbohydrate:protein (c:p) ratios. They show that c:p ratios affect female fly lifespan and fecundity depending on limiting dietary sterols. While the findings are mostly relevant to studies on dietary restriction in *Drosophila*, the findings do identify a caveat in our understanding of TOR-mediated longevity effects.

All three reviewers acknowledge the novelty and relevance of this work. I congratulate the authors for a nicely conceived and written work, which convincingly shows the key role of dietary sterols as limiting tradeoff/coupling units between reproduction and somatic maintenance when p:c ratios increase. This work also sheds light on how rapamycin impacts female fly lifespan on low sterol diet and high p:c ratios by indirectly preventing somatic sterol depletion via suppression of reproduction.

Title: The authors may consider changing the title to specify that dietary sterols are important for female flies responses to dietary restrictions.

Essential revisions:

1) The authors should address or at least extensively discuss how TOR signaling actually changes with diet and cholesterol, and whether that correlates with lifespan; for instance, do dTOR or dS6K mutant flies respond the same way to high protein/cholesterol diets?

2) The authors should discuss the following: a) Does lifespan and fecundity vary by calorie intake/energy density and not just P:C ratio? b) Could the lesson here be that cholesterol promotes survival on low calorie diets but not high calorie diets?

c) Maybe rapamycin can only promote longevity if the energy density/calorie consumption is below some threshold?

3) A discussion of how the documented effects of Rapamycin and Tor signaling in yeast, *C. elegans* and mammals relates to what is observed here is critical. The data imply that if egg production is shut down, Rapamycin won't have a longevity effect at all. It is unclear how to interpret that in relation to the effects in other organisms, where egg production is a much less energy / nutrient consuming proposition or where no eggs are produced (as in yeast).

4) To make the paper more accessible to readers not working with flies and highlight the more broadly relevant findings, Figures 2/3 and 4/5 could be combined into one or two figures altogether.

5) In the graphs of Figures 1-3, the authors need to clarify which data points are related to the lifespan axes and which ones to the egg production axes. For non-fly people, this is not obvious.

6) Lopez-Otin is mis-spelled in the first paragraph of the Introduction.

7) The authors should discuss bring up that while mammals should be able to survive by simply upregulating endogenous cholesterol synthesis, flies don't seem to be doing so (do they?). Or they do regulate cholesterol synthesis, is there is a difference between endogenous and dietary cholesterol?

---

## [Author Response]

[…] Title: The authors may consider changing the title to specify that dietary sterols are important for female flies responses to dietary restrictions.

We have taken your advice and specified this in the title.

Essential revisions:1) The authors should address or at least extensively discuss how TOR signaling actually changes with diet and cholesterol, and whether that correlates with lifespan; for instance, do dTOR or dS6K mutant flies respond the same way to high protein/cholesterol diets?

We agree that this is an important point of discussion. For cholesterol to be working through TOR to extend lifespan, we would expect that cholesterol addition should suppress TOR in a high protein environment. Current research indicates that this is not the case, and in fact, cholesterol has been recently identified as an up regulator of TOR. This has been discussed in –the subsection “Extending fly lifespan by dietary restriction involves an indirect trade-off”.

2) The authors should discuss the following: a) Does lifespan and fecundity vary by calorie intake/energy density and not just P:C ratio? b) Could the lesson here be that cholesterol promotes survival on low calorie diets but not high calorie diets? c) Maybe rapamycin can only promote longevity if the energy density/calorie consumption is below some threshold?

a) We have performed additional analyses which show that dietary calorie content does not have a significant effect on lifespan or egg production, but instead, these are explained by the P:C ratio and cholesterol contents. This has now been addressed in the Results section and two tables with details of the relevant statistics as well as a new supplementary figure have been added (Figure 2—figure supplement 2 and Supplementary files 3 and 4).

b) We have also included data to show that the benefits of cholesterol addition to egg laying and lifespan were not restricted to any pattern when considered across levels of dietary caloric density. This has now been addressed in the Results section and one supplementary figure has been added (Figure 2—figure supplement 2).

c) Rapamycin has already been shown to extend *Drosophila* lifespan across calorie densities – we now reference these data (subsection “Extending fly lifespan by dietary restriction involves an indirect trade-off”).

3) A discussion of how the documented effects of Rapamycin and Tor signaling in yeast, *C. elegans* and mammals relates to what is observed here is critical. The data imply that if egg production is shut down, Rapamycin won't have a longevity effect at all. It is unclear how to interpret that in relation to the effects in other organisms, where egg production is a much less energy / nutrient consuming proposition or where no eggs are produced (as in yeast).

We agreed that this point needed to be addressed and doing so has helped us to better communicate our findings. We have now clarified that the role of cholesterol and/or other micronutrients, and their role in mediating lifespan outcomes will be context specific. In particular, we have discussed how rapamycin may extend the lifespan of other organisms by suppressing growth/reproduction in yeast and mammals. We have also highlighted that rapamycin may rescue animals from the effects of protein toxicity, not just micronutrient deficiencies (subsection “Extending fly lifespan by dietary restriction involves an indirect trade-off”).

4) To make the paper more accessible to readers not working with flies and highlight the more broadly relevant findings, Figures 2/3 and 4/5 could be combined into one or two figures altogether.

This has been addressed and these four figures have now been condensed into two (Figures 2 and 3).

5) In the graphs of Figures 1-3, the authors need to clarify which data points are related to the lifespan axes and which ones to the egg production axes. For non-fly people, this is not obvious.

This has now been addressed by adding explanatory text to the figure legends of both Figures 2 and 3.

6) Lopez-Otin is mis-spelled in the first paragraph of the Introduction.

This has been amended.

7) The authors should discuss bring up that while mammals should be able to survive by simply upregulating endogenous cholesterol synthesis, flies don't seem to be doing so (do they?). Or they do regulate cholesterol synthesis, is there is a difference between endogenous and dietary cholesterol?

This has been addressed in the Discussion in conjunction with point 3. Here we have specified that mammals, unlike insects can synthesise their own cholesterol and then followed on to explain how the broader principle of micronutrient depletion might be relevant to higher organisms.